# Music Sound Quality Assessment in Bimodal Cochlear Implant Users—Toward Improved Hearing Aid Fitting

**DOI:** 10.3390/audiolres15060151

**Published:** 2025-11-06

**Authors:** Khaled H. A. Abdellatif, Horst Hessel, Moritz Wächtler, Verena Müller, Martin Walger, Hartmut Meister

**Affiliations:** 1Department of Otorhinolaryngology, Head and Neck Surgery, Medical Faculty, University of Cologne, Kerpenerstraße 62, D-50937 Cologne, Germanyverena.mueller@uk-koeln.de (V.M.); hartmut.meister@uni-koeln.de (H.M.); 2Jean-Uhrmacher-Institute for Clinical ENT-Research, University of Cologne, Geibelstr. 29-31, D-50931 Cologne, Germany; 3Cochlear Deutschland GmbH & Co. KG, D-30539 Hannover, Germany

**Keywords:** sound quality, cochlear implant, hearing aid, bimodal fitting, MUSHRA

## Abstract

**Background/Objectives**: Cochlear implants (CIs) are a common treatment of severe-to-profound hearing loss and provide reasonable speech understanding, at least in quiet situations. However, their limited spectro-temporal resolution restricts sound quality, which is especially crucial for music appraisal. Many CI recipients wear a hearing aid (HA) on the non-implanted ear (bimodal users), which may enhance music perception by adding acoustic fine structure cues. Since it is unclear how the HA should be fitted in conjunction with the CI to achieve optimal benefit, this study aimed to systematically vary HA fitting parameters and assess their impact on music sound quality in bimodal users. **Methods:** Thirteen bimodal CI recipients participated in a listening experiment using a master hearing aid that allowed controlled manipulation of HA settings. Participants evaluated three music excerpts (pop with vocals, pop without vocals, classical) using the multiple-stimulus with hidden reference and anchor (MUSHRA) test. To assess the reliability of individual judgments, each participant repeated the test, and responses were analyzed with the eGauge method. **Results:** Most participants provided reliable and consistent sound quality ratings. Compared to a standard DSL v5.0 prescriptive fitting, modifications in compression settings and low-frequency gain significantly influenced perceived music quality. The effect of low-frequency gain adjustments was especially pronounced for pop music with vocals, indicating stimulus-dependent benefits. **Conclusions:** The study demonstrates that HA fitting for bimodal CI users can be optimized beyond standard prescriptive rules to enhance music sound quality by increasing low-frequency gain, particularly for vocal-rich pieces. Additionally, the testing method shows promise for clinical application, enabling individualized HA adjustments based on patient-specific listening preferences, hence fostering personalized audiology care.

## 1. Introduction

Cochlear implantation allows partially restoring hearing in severe-to-profoundly-hearing-impaired individuals. The primary goal of cochlear implant (CI) provision is to improve speech recognition and hence to enable the listeners to participate in verbal communication. Indeed, many CI users reveal reasonable speech understanding, at least in quiet communication situations [1]. However, CI-mediated listening is far from the abilities a healthy auditory system allows for. Besides differences in dynamic range (see [2]), this is especially due to the fact that CIs, for example, typically provide a restricted spectro-temporal resolution, because of the limited number of electrodes that can be used to stimulate the auditory nerve (e.g., [3]). Additionally, interaction between the channels blurs the frequency mapping and temporal pitch cues are only of limited use. As a consequence, a CI mainly transmits envelope information as one of the features important for speech recognition (e.g., [4]). However, the lack of spectro-temporal fine structure may have consequences for other listening tasks.

In recent years, an increasing research effort has been directed towards CI users’ perception of non-speech sounds, especially music (e.g., [5]). For example, Akbulut et al. [6] applied a music-related quality of life questionnaire to compare adult CI recipients with a control group of normal-hearing listeners. They found that the importance of music was rated similarly in both groups despite CI users listening significantly less frequently to music. Moreover, the authors showed that music appraisal can have a strong negative impact on quality of life due to the restricted sound transmission. Additionally, the neural substrates of music perception [e.g., [7]] might be crucial due to cortical changes and plasticity associated with hearing loss and cochlear implantation.

Music is based on a number of different dimensions, such as rhythm, melody, harmony, and timbre. In terms of CI-mediated listening, several limitations arise in these dimensions (overview in [8]). Due to the pulsatile stimulation patterns of CI, temporal features of music such as rhythm are generally relatively well preserved. In contrast, the limited number of spectral channels restricts melodic tune identification [9], melody recognition [10] and perception of timbre [11]. The latter is related to different instruments and thus helps to distinguish multiple signals in music. It is also significantly connected to sound quality, which is itself a determinant of consumer satisfaction and quality of life [12,13]. Though there is no generally accepted definition of sound quality, it can be seen as a construct that is constituted by the acoustic properties of a stimulus and by the perceptual assessment, for instance in terms of accuracy or fidelity. Moreover, expectations might play a role with the assessment of sound quality. As such, sound quality is a highly subjective domain.

Given the apparent limitations of electric stimulation on music appraisal, a number of measures for improving sound quality have been discussed. For example, a number of electrode-mapping approaches have been suggested in order to render stimulation more appropriate to the needs of music transmission (e.g., [14]). Another way to tone down the negative consequences of limited spectro-temporal fine structure is the use of a hearing aid (HA) on the ear contralateral to the CI (bimodal stimulation) or even in combination with the CI on the implanted ear (hybrid stimulation). In these cases, the HA mainly transmits sounds from the low-frequency range where residual hearing is typically largest. Many CI recipients use bimodal stimulation that has positive consequences on speech perception (e.g., [15]). Beyond that, also music perception and sound quality may be improved [16].

However, it is still unclear how the HA should be fitted in order to achieve the best sound quality in bimodal patients. HAs are typically adjusted using generic fitting rules such as DSL v5.0 (desired sensation level [17]), NAL-NL2 [18], or manufacturer-specific approaches (e.g., [19]). These rules use patient-specific data, typically from the pure-tone audiogram, as input and calculate prescriptive gain as a function of frequency, level, and hearing loss for long-term average speech spectrum signals. However, there is currently no generally accepted fitting rule for a HA in combination with the CI in order to maximize perception. In the framework of a survey on audiologists, Scherf & Arnold [20] found many of them reported that the HA is not re-fitted when used in conjunction with a CI. This often has practical reasons since CI and HA provision may take place at different institutions. Others reported that at least a loudness balancing between the two devices is performed, as suggested by Ching et al. [21]. However, a structured procedure to consider the apparent differences between the CI and HA, which have to be taken into account for bimodal hearing, obviously does not exist yet.

Recently, Vroegop et al. [19] tapped into this issue by comparing three different approaches of hearing aid fitting in the framework of bimodal listening. They considered two different fitting formulas as well as a loudness balancing based on broadband and narrowband signals. The different procedures were assessed by means of localization and speech recognition, but in general, no clear differences were found. Once sufficient gain is provided by the HA, it may be the case that further fine-tuning in conjunction with the CI does not significantly increase the outcome. However, it must be kept in mind that the study focused on performance measures and that the findings might be different when subjective criteria, such as (music) sound quality, are considered.

The present study aims at addressing this aspect by systematically assessing the effects of different HA adjustments on sound quality ratings in bimodal listening. Here, the rationale is to manipulate only the HA and use the CI in its regular, optimally fitted condition. In order to keep the factors controllable, a “master hearing aid” (openMHA 4.17.0 [22]) was used instead of real hearing aids. Three different excerpts of music were considered and transmission via the MHA was manipulated in terms of frequency range, compression, and low-frequency gain. The ultimate goal of this study is to give information for a possible modification of HA fitting procedures in terms of improving sound quality. As the outcome is based on subjective ratings, the reliability of the individuals’ quality estimations is a crucial aspect of reaching this goal and was thus an important element of the study.

## 2. Materials and Methods

### 2.1. Subjects

Thirteen bimodally fitted listeners (three female and ten male) with a mean age of 55 years (range 28–71 years, SD = 17.2) were recruited in this study (see Table 1 for details). They were all provided with CochlearTM sound processors CP910 (n = 1) and CP1000 (n = 12). This restricted choice of devices was used to keep the methodological and especially the technical variability small. As this was the main inclusion criterion for the study, this resulted in a relatively low sample size and large variability in terms of hearing loss etiology or musical experience. Residual hearing of the non-implanted ear was mainly present in the frequency range up to 1 kHz. Subjects with a hearing loss of not larger than 85 dB HL in this frequency range were chosen, as no benefit in music perception is expected for thresholds exceeding this value [23]. The corresponding low-frequency average (250–1000 Hz) was between 38.3 and 83.7 dB HL.

All participants were native German speakers and had normal or corrected-to-normal vision. Prior to the experiment, they were given detailed information about the study and informed consent was obtained. Participants were reimbursed with EUR 10,-/h. The study protocol was approved by the local ethics committee (reference 18-383, approved 13 February 2019).

### 2.2. Setup

The setup is based on a computer (Samsung Ultrabook Model: NP530U4E running Windows 8) where the stimuli are generated and the participant’s responses are collected. Since the rationale was to modify hearing aid fitting while leaving the cochlear implant untouched, the original stimulus was transmitted to the CI. This was accomplished by connecting a soundcard (MAYA44USB, ESI Audiotechnik GmbH, Leonberg, Germany) to the Cochlear™ Wireless TV Streamer and transmitting via Bluetooth to the sound processor. This transmission line caused a delay of 40 ms relative to the acoustic stimulation of the non-implanted ear, which was compensated for by delaying the contralateral stimulus accordingly. The acoustic stimulation was conducted via the open Master Hearing Aid (openMHA [22]). Using the openMHA instead of a real hearing aid was essential for standardization of signal transmission and considers basic aspects such as frequency-dependent amplification and dynamic range compression (DRC). A 10-channel multiband dynamic compression with center frequencies ranging from 250 Hz to 8000 Hz was implemented by the openMHA. Based on the individual pure-tone audiogram, frequency gain shaping and dynamic range compression were prescribed for each listener using DSL v5.0 gains for 50, 65, and 80 dB SPL input levels using the Otometrics OTOsuite software release date 25.06.2020 (Natus Medical Denmark ApS, Taastrup, Denmark). Linear inter-/extrapolation was used to obtain gains for input levels ranging from 0 to 120 dB SPL in steps of 1 dB. Gains were limited so that the resulting predicted real-ear levels did not exceed the OSPL90 values provided by DSL v5.0. Gain values were manually inputted as a gain table into the openMHA software. The fast-acting compression attack and release times were 0.02 and 0.1 s, respectively. The audio output of the soundcard was connected to a headphone amplifier (Superlux HA3D, Superlux Enterprise Development (Shanghai) Co., Ltd., Shanghai, China) and transmitted on a hearing aid receiver (SureFit2 (HP), ReSoundGN, Mumbai, India) coupled to the ear canal by a tympanometry tip and enclosed in an impression material. This closed fitting ensured that no acoustic leakage occurred, which may severely reduce low-frequency energy.

### 2.3. Music Stimuli

Three excerpts of different music pieces were included in the assessment (duration 14 s each). Pop music with vocals was represented by an excerpt of Namika’s “Je ne parle pas francais” including a standard instrumentation (guitar, keyboard, bass, and drums), and a female vocal. The same excerpt was considered as a pure instrumental version in order to examine the effect of vocals on sound quality. Moreover, classical music was covered by an excerpt of Johann Strauss’ “The blue danube” played by a full orchestra, representing a complex instrumentation. These excerpts were chosen to give different genres and different acoustic characteristics (i.e., rhythm, timbre, complexity, vocals). They were also chosen in order to represent a reasonable amount of acoustic information in the low-frequency range (i.e., below 1 kHz) where residual hearing typically plays the most important role. To this end, the spectral centroid was calculated for each music piece using the MATLAB MIRtoolbox version 1.10.0.0 [24]. The average spectral centroid across the duration of the music pieces was 683 Hz for classical, 613.8 Hz for pop with vocals, and 383 Hz for pop without vocals. Shorter parts of these three recordings (duration 2.5 s) were additionally used to assess just-noticeable differences (jnds) regarding changes in low-frequency gain. These additional measurements were performed to ensure that the modifications of the stimuli in the MUSHRA assessment (see below) were actually perceivable by the listeners.

### 2.4. Sound Quality Assessment

The “multiple-stimulus with hidden reference and anchor” (MUSHRA) test was used for sound quality assessment (see recommendation [25]). This method does not require an overt reference such as the healthy ear in patients with single-sided deafness [26]. Instead, it allows simultaneous comparison and rating of different stimuli. Typically, the stimuli are compared against a known reference stimulus that is chosen to have the best quality. However, in our case the reference (i.e., fitting based on DSL v5.0) was only presented in a hidden form, since it must not necessarily present optimal sound quality. The anchor, which should represent poor quality, was generated by a noise vocoder [27]. The MATLAB code is available online (see [28]). The signal was filtered into two bands having equally spaced boundaries based on a 35 mm basilar membrane distance [29] across a frequency range between 0.2 and 20 kHz. Then the output of each band was half-wave-rectified and low-pass-filtered at 250 Hz to extract the temporal envelope. This was then multiplied by a wide-band noise carrier, and the resulting signal was summed across the bands. Rating was performed on a 100 point scale with semantic labeling of sound quality (i.e., “bad” (range 0–20 pts), “poor” (20–40 pts), “fair” (40–60 pts), “good” (60–80 pts), and “excellent” (80–100 pts)). This way, the “mean opinion scores” (MOSs) were derived. The MUSHRA “drag and drop” version [30], which represents a more intuitive user interface, was applied for the sound quality assessment (see Figure 1). This version allows activating the stimulus by clicking on the corresponding button (always labeled A-G for the different stimuli, with the hidden reference presented twice). The next step is to drag the button into the rating field and to drop it onto the appropriate rating position. All stimuli could be played and dragged as often as desired until a final decision about the sound quality was reached.

### 2.5. Procedures

After the participants were instructed about the study rationale, informed consent was obtained. The sound quality assessment based on MUSHRA was conducted for two conditions, a “general” and a “fitting” condition, each with the DSL-based gain adjustments as the hidden reference presented twice in each condition. The “general condition” considered more basic aspects of bimodal listening, such as sound quality for each device alone, linear vs. non-linear amplification, and overlap vs. non-overlap of the frequencies of the cochlear implant and the hearing aid (i.e., HA ≤ 1 kHz, CI > 1 kHz). In the latter case, the original stimulus was not sent to the CI but rather a low-pass filtered version with a cutoff frequency of 1 kHz, approximating the upper limit of the main frequency range transmitted by the openMHA. The no-overlap condition was motivated by the fact that some literature pointed to a decrease in music perception when CI and HA share the same frequency region (e.g., [31]).

The “fitting condition” considered gain manipulations of the HA in the low-frequency range ≤ 1 kHz. Gain was either decreased or increased by one or two individual steps. Individual stepsize was used to ensure that gain manipulations were always perceivable. Therefore, just-noticeable differences (jnds) were assessed in a 3I3AFC paradigm [32] presented via the openMHA. Jnds associated with 75% correct discrimination were determined for changes in low-frequency gain. Stepsize 1 was set to 1.5 jnd, consequently approximating differences that were small but detectable. Individual stepsize averaged across the three music excerpts is shown in Table 1. Stepsize 2 represented the double value of step 1. For large jnds, the stepsize might cause clipping. However, in this case, stepsize 2 was limited to the maximum gain value before any distortions occurred. Table 2 summarizes the conditions.

Prior to the sound quality ratings, individual intensity adjustments to reach the most comfortable loudness level were carried out for each music piece. This was performed for the CI and the HA alone as well as for the combination of both devices. After loudness adjustment, the MUSHRA concept was explained by giving examples for the rating procedure. Half of the listeners began quality assessment in the general condition and half in the fitting condition. The presentation of the different music excerpts was pseudo-randomized. After a break of 15 min a complete retest was performed in order to check reliability. In total, each listener gave 84 assessments (3 music pieces × 7 processing × 2 conditions × 2 tests) using the MUSHRA procedure. Total testing time was about 2 h.

Reliability of the ratings was assessed by eGauge [33]. This method calculates the reliability using a non-parametric permutation test [34] as a test of significance. The permutation test is computed using several iterations per subject and defines the noise floor of the performance for reliability metrics. Below that level, the performance is equivalent to random ratings that could degrade quality of data. Consequently, sound quality assessments where the listeners did not fulfil this criterion were not considered for further analysis (see Table 1).

### 2.6. Statistical Analysis

Data were subjected to a one-way repeated-measures analysis of variance (rmANOVA) with processing condition as within-subject factor. Greenhouse–Geisser correction was applied if the assumption of sphericity was violated. Paired comparisons with Bonferroni correction were conducted as post hoc analyses. The *p*-value was set to 0.05. Analyses were performed with IBM SPSS statistics v. 27.

## 3. Results

### 3.1. General Condition

Figure 2 shows the results of the “general condition” for the excerpt of classical music (panel A), pop with vocals (panel B), and pop without vocals (panel C). Panels A and B show relatively similar patterns, with the DSL reference revealing the best and the anchor revealing the worst sound quality. However, panel C appears to be different since all conditions show relatively similar outcomes and the variability of the MOS is higher than for the other two music pieces.

An rmANOVA on the MOS-data of the classical music excerpt with processing condition as the independent variable shows a significant main effect of processing (F(1.721, 5) = 9.6, *p* = 0.002, ƞ_p_^2^ = 0.49). Post hoc pairwise comparisons with Bonferroni correction reveal a significantly higher MOS for DSL compared to all other processing conditions (*p* ≤ 0.031). Moreover, MOS for the anchor stimulus reveals significantly lower sound quality than both, DSL (*p* = 0.011) and linear fitting (*p* = 0.003).

An rmANOVA on the MOS-data of the pop music excerpt with processing condition as the independent variable also shows a significant main effect of processing (F(2.562, 5) = 8.375, *p* = 0.001, ƞ_p_^2^ = 0.511). Post hoc pairwise comparisons with Bonferroni correction reveal a significantly higher MOS for DSL compared to the CI alone (*p* = 0.010) and the HA alone (*p* = 0.025) whereas DSL is not significantly different from the no-overlap setting (*p* = 0.063) and the linear gain condition (*p* = 0.081), based on the chosen significance level. The anchor stimulus is rated significantly poorer than the DSL reference(*p* = 0.005).

Lastly, an rmANOVA on the MOS-data of the pop music excerpt without vocals with processing condition as the independent variable again shows a significant main effect of processing (F(1, 5) = 4.168, *p* = 0.003, ƞ_p_^2^ = 0.294). Post hoc pairwise comparisons with Bonferroni correction reveal a significantly higher MOS for DSL compared to HA alone (*p* = 0.010) and the no-overlap condition (*p* = 0.015). However, no further significant differences can be found. Notably, the anchor stimulus is not rated significantly worse than any of the other processing conditions.

Taken together, both the classical and pop music excerpts show the highest sound quality for the DSL reference and the lowest sound quality for the anchor stimulus, as intended. Listening with the CI and the HA alone yields significantly worse sound quality than the combination of both devices. However, this does not apply for the no-overlap condition that yields MOSs in the range of the single devices. The linear processing condition is also not significantly different than the two devices separately but reveals higher MOSs, at least in absolute terms.

The pop music excerpt without vocals appears to cause somewhat different results, since the conditions revealed more similar results and the anchor did not yield the intended low sound quality. In fact, despite an absolute difference of about 15 points, the anchor was not rated significantly worse than the DSL reference.

### 3.2. Fitting Condition

Figure 3 shows the results of the “fitting condition” for the excerpt of classical music (panel A), pop with vocals (panel B), and pop without vocals (panel C). Again, the DSL reference mostly yields highest sound quality but changing low-frequency gain appears to affect the ratings.

A rmANOVA on the MOS-data of the classical music excerpt with processing condition as the independent variable shows a significant main effect of processing (F(2.306, 5) = 6.2, *p* = 0.005, ƞ_p_^2^ = 0.383). Post hoc pairwise comparisons with Bonferroni correction reveal a significantly higher MOS for DSL compared to the anchor stimulus (*p* = 0.004) as well as gain reductions of one (*p* = 0.003) and two steps (*p* = 0.007).

A rmANOVA on the MOS-data of the pop music excerpt with processing condition as the independent variable shows a significant main effect of processing (F(1, 5) = 27.715, *p* < 0.001, ƞ_p_^2^ = 0.776). Post hoc pairwise comparisons with Bonferroni correction reveal a significantly higher MOS for DSL compared to the anchor stimulus (*p* = 0.004) and a gain reduction of two steps (*p* = 0.018). However, a significantly higher MOS for a gain increase of 2 steps compared to the DSL reference can also be observed (*p* = 0.017).

In contrast, a rmANOVA on the MOS-data of the pop music excerpt without vocals with processing condition as the independent variable does not show a significant effect of processing (F(1.641, 5) = 0.649, *p* = 0.506, ƞ_p_^2^ = 0.061).

Basically, this analysis revealed that manipulating low-frequency gain affects sound quality assessments. Still, the effect seems to depend on the piece of music presented. Whereas for the classical excerpt only a decrease in sound quality with decreasing gain relative to the DSL reference could be found, a better sound quality with increasing gain was additionally observed for the pop music excerpt with vocals. However, when this piece of music was presented in an instrumental version, no significant effect of gain manipulation could be shown.

## 4. Discussion

This study focused on music sound quality in bimodal CI users by manipulating the settings of a (master) hearing aid and determining the corresponding effects by using a multiple-stimulus (MUSHRA) method. The overall goal of the study was to suggest possible modifications in hearing aid fitting that might improve subjectively perceived sound quality.

The study considered two different conditions. One condition tapped into more general aspects of bimodal fitting such as the sound quality perceived with either device alone, the overlap of the frequency regions transmitted by the CI and the HA, and a linear amplification, compared to the compressive standard DSL fitting. The second condition addressed a more direct fitting aspect, namely the decrease or increase in low-frequency gain of the hearing aid. The fundamental study concept was that any manipulation was performed on the hearing aid while keeping the settings of the cochlear implant as used by the individual listener.

Three different music excerpts were selected. Their choice was based on different considerations. First, in order to reflect manipulations of the HA that typically provides low-frequency gain in bimodal CI users, a significant proportion of the spectrum should lie in this frequency area. This was ensured by determining the spectral centroid of the stimuli that was below 1 kHz in all cases, i.e., 683 Hz for classical, 613.8 Hz for pop with vocals, and 383 Hz for pop without vocals. Second, the music excerpts should represent differently complex stimuli by taking a piece with a rich instrumentation (classical), and a more simple instrumentation with and without vocals into account (pop). The fact that vocals appear to play a unique role for CI recipients (e.g., [35]) was taken into account by using the same excerpt of a single song, once with and once without vocals.

Sound quality judgements were assessed using the multiple-stimulus with hidden reference and anchor test (MUSHRA), typically denoted as CI-MUSHRA when used with cochlear implant recipients [36]. We used the “drag and drop” version of the MUSHRA that has practical advantages as it presents a more intuitive user interface [30].

### 4.1. General Condition

The general condition revealed that the reference fitting based on DSL provided the highest sound quality. Using only the CI or the hearing aid significantly decreased MOS, as expected, and in line with D’Allessandro et al. [37], who showed a significant reduction in overall sound quality when the HA was removed and the ear was plugged. There was no significant difference between using the two devices in isolation. Since MOS coarsely doubled in the combined reference condition, this might give evidence that the CI and the HA largely present complementary sound cues [38]. Hence, both the envelope information over a broader frequency range, predominantly transmitted by the CI, and the acoustic fine structure cues in the low-frequency area transmitted by the HA contribute to music sound quality.

The no-overlap condition was considered to control whether sharing the same frequency range (i.e., below 1 kHz) is detrimental, as discussed in the literature. Though not conclusive, some studies reveal negative effects of overlap and non-optimal frequency-to-place coding with bimodal stimulation (e.g., [31,39]). However, we found that in the “no-overlap” condition sound quality clearly decreased. This could be due to several reasons. First, it is unclear whether a changed frequency allocation of the CI caused by the low-pass filtered stimuli impacted ratings negatively. Another factor could have been that the “no-overlap” condition disturbs the impression of “stereo listening”, which is important for sound quality, as recently shown by Landsberger et al. [26].

Finally, we also included a condition based on DSL, but with a linear sound processing instead of the dynamic compression. This condition somewhat decreased perceived sound quality, although the difference to the DSL setting was not statistically significant for all music excerpts. The influence of compression on sound quality ratings is inconclusive. Some studies have shown that compressive is preferred over linear fitting. For instance, Gilbert et al. [40] manipulated the “maplaw” (i.e., the mapping of the acoustic input to the electric dynamic range) as well as the automatic gain control (AGC) in Med-El CI-processors. Whereas AGC had only subtle effects on perceived sound quality, higher maplaw (i.e., more compression) was associated with better quality ratings. Subjective feedback of the listeners revealed better audibility for soft sounds and lyrics and basically a “fuller” impression of the sound as a possible reason. In contrast, Kirchberger & Russo [41] showed, in hearing aid users with predominantly moderate hearing loss, that linear was preferred over compressive processing with different compression ratios. However, results from hearing aid users might not be directly comparable with bimodal cochlear implant users as different frequency regions are transmitted and the tradeoff between increased audibility of soft sounds and possible distortions caused by compression may be different.

These results apply to the classical and pop music excerpts with vocals in a similar manner. However, notably, the pop music piece without vocals showed somewhat unusual results in that differences between the various conditions were clearly smaller. In particular, the anchor stimulus revealed a relatively high rating that was not seen for the two other music excerpts. This suggests that for the sparsely instrumented stimulus without vocals, different cues are at work. We speculate that mainly rhythmic aspects played a role here, which were less sensitive to the signal manipulations applied—including the noise vocoding performed to generate the anchor stimulus.

### 4.2. Fitting Condition

The fitting condition specifically addressed changes in low-frequency gain. Low frequencies play an important role for sound quality; however, this frequency area might be critical for CI users due to a potential lack of stimulation in the apical cochlear region [36,42,43]. Hence, complementary transmission of this information by the hearing aid might be especially crucial.

The fitting condition showed significant effects of low-frequency gain manipulations, similar to effects of frequency response manipulations in cochlear implants on sound quality [44]. Compared to the DSL reference, decreasing the low-frequency gain yielded a significantly lower sound quality in both the classical and pop music excerpts. We assume that reducing low frequencies makes the sound “thinner”, hence negatively affecting perceived quality. Moreover, increasing low-frequency gain significantly increased sound quality relative to DSL, but only for pop music with vocals. This is consistent with the proposed special role of vocals for sound quality in listeners with CI [35]. For instance, Gfeller et al. [45] stated that “when musical stimuli have lyrics, some factors that have been shown to influence speech perception also influence accuracy of music perception”. Moreover, Buyens et al. [46] asked normal-hearing listeners and cochlear implant recipients to mix the tracks of different music pieces in order to generate a version that sounded most enjoyable. The CI users generated mixes that showed a relatively more intensive adjustment of the voices. In line with this is the recent study by Limb et al. [47], who showed that music was preferred by CI users when the voice of the sound was amplified by 9 dB—a value that is in the range of the gain-steps adjusted in the present study (see Table 1).

The improvement of sound quality due to increased low-frequency gain may not necessarily be restricted to cochlear implant recipients. Vaisberg et al. [48] asked listeners with mild–moderate hearing loss to adjust gain settings as individually preferred. Similarly to our study, they used the openMHA as a hearing aid simulation. In comparison to the standard DSL-fitting, the largest adjustments of low-frequency gain were found in pop music, followed by classical music. The authors concluded that a standard fitting might be appropriate in terms of speech recognition but not necessarily when listening to music.

Taken together, our study provides some evidence for an improvement in music sound quality in bimodally fitted CI users by manipulating the low-frequency acoustic transmission via the hearing aid. First, linear amplification appeared to be inferior to dynamic compression as applied based on the standard DSL fitting. This was found for all of the three music excepts though the difference was not always statistically significant—possibly due to the large inter-individual variance in sound quality ratings. Secondly, sufficient low-frequency gain—as proposed by DSL—is important. Otherwise, sound quality is compromised, as was shown with decreasing gain for both pop with vocals and the classical piece. Furthermore, sound quality was improved by increasing the low-frequency gain in the pop music excerpt with vocals. Consistent with other research, this supports the extra role of vocals for music perception with CI. In contrast, gain manipulations in sparsely instrumented music, such as the piece of pop without vocals, showed a much smaller effect. We suspect that this is due to the superior role of rhythmic cues, which might dominate for this type of instrumentation. Thus, when trying to improve sound quality by modifying hearing aid fitting in bimodal cochlear implant recipients, different types of music need to be considered.

## 5. Conclusions

This study showed that increasing the acoustic low-frequency gain has the potential to improve sound quality, especially for music with vocals. It thus points to possible clinical applications, e.g., by increasing hearing aid amplification for bimodal fitting, based on standard rules such as DSL, in a music-specific fashion, or by integrating higher low-frequency amplification into the fitting software in an application-specific manner. Methodologically, the openMHA was used in order to provide the acoustic path to the listener. We opted for the openMHA instead of the individual hearing aid of the listener because it offers straightforward access to the relevant parameters (in our case, compression and frequency-dependent gain) and manipulations can be performed in an arbitrary manner. This would not have been readily feasible in real hearing aids where (unknown) signal processing algorithms may have been at work. For a similar reason we decided to take only two types of speech processors from the same manufacturer into account, which also helped to keep methodological variability low. Another strength of the study is the assessment of the reliability of the individuals’ answers. Using eGauge, it could be shown that most but not all of the subjects provided reliable ratings. To the best of our knowledge, this is the first study that explicitly determined the reliability of sound quality ratings in CI users.

A limitation is the small study sample that was mainly caused by the restricted choice of processor types, as explained above. This allowed a more controlled study setup on the one hand but limits the generalizability of the results on the other hand as it is not entirely clear how the (simulated) hearing aid would have interplayed with other electric stimulation strategies. For instance, assuming that electric stimulation could provide more low-frequency information, the effect of modifying acoustic low-frequency gain may be weakened. Moreover, the limited sample size made it particularly difficult to perform a subgroup analysis, e.g., based on the residual hearing of the participants. In addition, as a first step, the aim was to address more fundamental aspects of hearing aid fitting, recognizing that more subtle modifications (e.g., in terms of compressor speed, cf. [49]) could have additional effects on sound quality. Another restriction was the choice of a relatively small sample of music excerpts, which, however, were specifically selected, and the use of the openMHA instead of a real hearing aid, which might limit transfer of the results to real-life conditions. This latter point is currently being addressed in a follow-up study. Last but not least, as an acute trial, this study does not consider long-term adaptation effects, which could be a focus of longitudinal studies.

## Figures and Tables

**Figure 1 audiolres-15-00151-f001:**
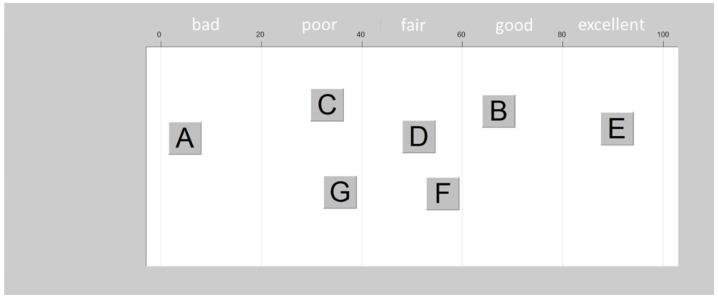
Schematic of the MUSHRA drag and drop interface. Each button depicted by a letter represents a sound sample. The letter labels (A–G) are arbitrary and randomized. The participants evaluated these stimuli presented through both the CI and HA by positioning the buttons along the quality scale between 0 and 100.

**Figure 2 audiolres-15-00151-f002:**
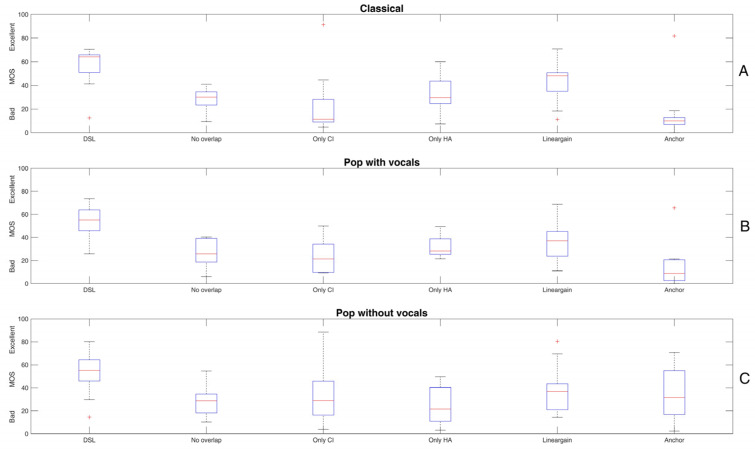
Mean opinion scores (MOSs) for the excerpt of classical music (**A**), pop music with vocals (**B**) and pop without vocals (**C**). The conditions displayed on the *x*-axis are given in Table 2. The red line shows the median, and “+” symbols indicate outliers. Typically, DSL yields highest MOS and the anchor lowest MOS, depending on the music excerpt. Please refer to the text for the statistical analyses.

**Figure 3 audiolres-15-00151-f003:**
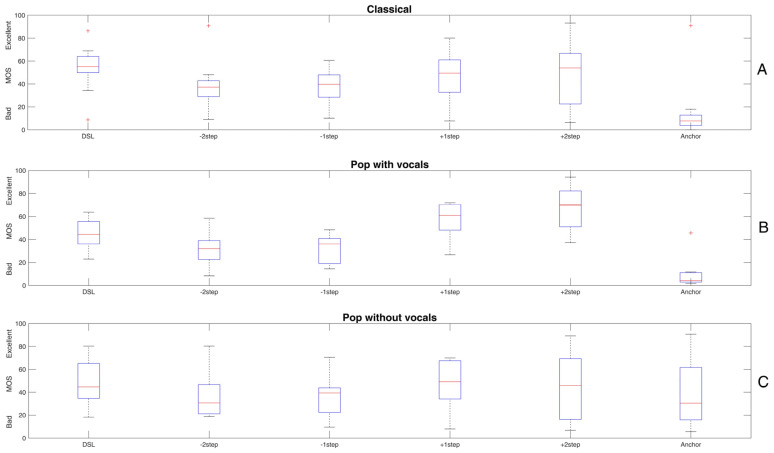
Mean opinion scores (MOS) for the excerpt of classical music (**A**), pop music with vocals (**B**), and pop without vocals (**C**). The conditions displayed on the *x*-axis are given in Table 2. The red line shows the median, and “+” symbols indicate outliers. Increasing low-frequency gain tends to yield higher MOS, depending on the music excerpt. Please refer to the text for the statistical analyses.

**Table 1 audiolres-15-00151-t001:** Subject characteristics. Low-frequency pure-tone average (LF PTA) was calculated across 250, 500, and 1000 Hz. LF gain stepsize (averaged across the three music excerpts) displays the value derived from the just-noticeable differences (jnds) and taken as stepsize 1. The three right columns show the reliability (Reliab.) of the data as assessed by the eGauge method. For reasons of anonymity, an age range is given.

ID	Gender	Age Range [yrs]	CI Experience [yrs]	HA Experience [yrs]	LF PTA [dB HL]	LF Gain Stepsize [dB]	Reliab. Classical	Reliab. Pop with Vocals	Reliab. Pop No Vocals
S01	f	71–75	5	20	70.0	5.1	yes	yes	yes
S02	m	56–60	4	4	63.0	7.1	no	yes	yes
S03	m	51–55	1.5	3	43.7	6.0	yes	yes	yes
S04	m	66–70	1.6	55	82.3	7.2	yes	yes	no
S05	m	51–55	1.1	27	39.3	4.5	yes	yes	yes
S06	m	26–30	1.1	20	83.7	5.8	yes	yes	yes
S07	f	76–80	1.0	11	38.3	12.6	no	no	no
S08	m	26–30	6.0	16	76.0	7.0	yes	yes	yes
S09	m	66–70	1.0	6	81.0	11.3	yes	no	yes
S10	m	66–70	0.3	16	44.7	6.3	yes	yes	yes
S11	f	36–40	0.6	16	55.3	8.0	yes	yes	yes
S12	m	36–40	0.3	31	80.0	6.2	yes	no	yes
S13	m	61–65	2	26	53.0	10.7	yes	no	yes

**Table 2 audiolres-15-00151-t002:** Description of the different stimuli and manipulations used in the “general condition” and “fitting condition”.

**General Condition**	
*Condition*	*Description*
DSL	Standard fitting based on DSL v5.0
No overlap	Electric and acoustic stimulation do not share a common frequency range
Only CI	Stimulation via MHA turned off
Only HA	Stimulation via CI turned off
Linear gain	Compression ratio of the MHA set to 1:1
Anchor	Noise-vocoded signal as a reference for poor sound quality
**Fitting Condition**	
*Condition*	*Description*
DSL	Standard fitting based on DSL v5.0
−2 step	Low-frequency gain reduced by 2 individual steps
−1 step	Low-frequency gain reduced by 1 individual step
+1 step	Low-frequency gain increased by 1 individual step
+2 step	Low-frequency gain increased by 2 individual steps
Anchor	Noise-vocoded signal as a reference for poor sound quality

## Data Availability

The original data presented in this study are available on reasonable request from the corresponding author. The data are not publicly available due to privacy concerns.

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
