# Peer review of "Music Sound Quality Assessment in Bimodal Cochlear Implant Users—Toward Improved Hearing Aid Fitting"

_audiolres, 2025, doi:10.3390/audiolres15060151_

Round 1
Reviewer 1 Report
Comments and Suggestions for Authors
The authors present an important topic-- optimizing acoustic stimulation parameters for patients in bimodal condition focusing on sound quality. While the background content is well presented overall, I think the details about the manipulations could be explained more clearly and shared in a Figure and or Table. I would appreciate seeing clearly what were the different conditions tested and clear depiction of the stimuli presented to patients.
Another important limitation of the approach used for this population needs more discussion is inherent to using MUSHRA in patients with hearing loss. Particularlry using CI but also relevant to this study is that there is no "normal hearing ear" for the reference. Landsberger, et al (cited in the references) avoids this issue as they tested SSD patients and used the normal hearing ear as the "ruler." In this study, some of the patients have significant hearing loss (severe range) in their acoustic ear.
I think consideration and some discussion about this is important-- did these individuals with significant hearing loss truly identify the reference as "100", the highest quality rating to which compare all other samples? This is critical to use this approach effectively and is not valid when presenting to the reference to a CI ear, for example.
minor grammatical point:
line 239 "peace" should be "piece"
Author Response
Comments 1: The authors present an important topic-- optimizing acoustic stimulation parameters for patients in bimodal condition focusing on sound quality. While the background content is well presented overall, I think the details about the manipulations could be explained more clearly and shared in a Figure and or Table. I would appreciate seeing clearly what were the different conditions tested and clear depiction of the stimuli presented to patients.
Response 1: Thank you very much, a table (#2) describing he conditions and stimuli has now been included.
Comments 2: Another important limitation of the approach used for this population needs more discussion is inherent to using MUSHRA in patients with hearing loss. Particularlry using CI but also relevant to this study is that there is no "normal hearing ear" for the reference. Landsberger, et al (cited in the references) avoids this issue as they tested SSD patients and used the normal hearing ear as the "ruler." In this study, some of the patients have significant hearing loss (severe range) in their acoustic ear.
Response 2: Thank you for raising this important point. In order to make clearer that MUSHRA does not need an overt reference (such as the normal hearing ear) but that the quality assessments are based on the comparisons of the various conditions, we write: “The “multiple-stimulus with hidden reference and anchor” (MUSHRA) test was used for sound quality assessment (see recommendation [25]). This method does not require an overt reference such as the healthy ear in patients with single sided deafness [26 –Landsberger et al. 2020]. Instead, it allows simultaneous comparison and rating of different stimuli. Typically, the stimuli are compared against a known reference stimulus that is chosen to have the best quality. However, in our case the reference (i.e., fitting based on DSL v5.0) was only presented in a hidden form, since it must not necessarily present optimal sound quality.”
In general, MUSHRA has been used several times in patients with hearing loss and/or cochlear implants – sometimes even labelled as CI-MUSHRA ([42] Roy et al. 2012, DOI: 10.1177/1084713812465493)
Comments 3: I think consideration and some discussion about this is important-- did these individuals with significant hearing loss truly identify the reference as "100", the highest quality rating to which compare all other samples? This is critical to use this approach effectively and is not valid when presenting to the reference to a CI ear, for example.
Response 3: In the MUSHRA-approach, the reference is hidden. It must not necessarily yield the highest sound quality (i.e., 100). To avoid this potential misunderstanding we have reworded the text (please see above).
minor grammatical point:
Comments 4: line 239 "peace" should be "piece"
Response 4: Thank you, this was corrected.
Reviewer 2 Report
Comments and Suggestions for Authors
Overall, this paper is a great demonstration of the authors' work. A few limitations to be considered: 1) the overall population is too small, 2) the average CI wearing time for the research population is varied/seems to have a high standard deviation. I wonder if the results could be different if the wearing time can be more similar across the population.
Author Response
Comments 1: Overall, this paper is a great demonstration of the authors' work. A few limitations to be considered: 1) the overall population is too small, 2) the average CI wearing time for the research population is varied/seems to have a high standard deviation. I wonder if the results could be different if the wearing time can be more similar across the population.
Response 1: Thank you very much for your comment. We acknowledge this in the methods section: “They were all provided with CochlearTM sound processors CP910 (n=1) and CP1000 (n=12). This restricted choice of devices was due to keep the methodological and especially the technical variability small. As this was the main inclusion criterion for the study, this resulted in a relatively low sample size and large variability in terms of hearing loss etiology or musical experience.”
We also have extended the description of potential limitations under “Conclusions”. We write: “A limitation is the small study sample that was mainly caused by the restricted choice of processor types, as explained above. This allowed a more controlled study setup on the one hand but limits the generalizability of the results on the other hand as it is not entirely clear how the (simulated) hearing aid would have interplayed with other electric stimulation strategies. For instance, given that electric stimulation could provide more low frequency information the effect of modifying acoustic low frequency gain might be weakened. Moreover, the limited sample size made it particularly difficult to perform a subgroup-analysis, e.g. based on the residual hearing of the participants.” (Please see also the comments of review 3).
Reviewer 3 Report
Comments and Suggestions for Authors
General Comments
The study demonstrates novelty in exploring HA parameter adjustments for music sound quality in bimodal CI users, a area with limited research. The integration of dynamic fitting modifications (e.g., compression settings and low-frequency gain) with subjective assessments via MUSHRA is a strength. However, the small sample size (n=13) and focus on a single CI manufacturer (Cochlear devices) may limit the generalizability of findings. I encourage author to acknowledge these limitations more explicitly and discuss their implications in the discussion section. Additionally, while the methodology is sound, some aspects of presentation and analysis could be refined to improve clarity and depth.
Specific Comments
- Abstract
- The abstract provides a good overview but could better highlight the key findings and their clinical significance. Specifically, emphasize that modifications to HA fitting (e.g., increased low-frequency gain for vocal-rich music) can significantly improve sound quality, as this is a central contribution. Also, mention the practical implications for personalized audiology care.
- Introduction
- The introduction sets the context well, but it would benefit from a more focused rationale on why music sound quality is a critical outcome measure for bimodal users. Consider citing recent studies (e.g., from the last 2-3 years) on music perception in CI users to update the literature review and strengthen the motivation. For example, work on neural correlates of music appreciation or patient-reported outcomes could be added.
- Methods
- Participants: The sample characteristics are described, but please provide more detail on the recruitment process and any inclusion/exclusion criteria related to musical experience or hearing loss etiology. This will help readers assess the representativeness of the cohort.
- Procedure: The use of the MUSHRA test with the drag-and-drop interface is well-explained. However, clarify how the "general condition" and "fitting condition" were randomized across participants to avoid order effects. Also, mention any training or familiarization sessions provided to participants to ensure reliable ratings.
- Stimuli: The music excerpts are appropriately chosen, but include the spectral centroid values for each excerpt in the main text (not just in the methods) to better justify their selection for low-frequency emphasis. For instance, state: "The average spectral centroid was 683 Hz for classical, 613.8 Hz for pop with vocals, and 383 Hz for pop without vocals, ensuring sufficient low-frequency content."
- Figure 1: The MUSHRA interface is clearly depicted, but the caption could be expanded to explain how this interface was used in the context of bimodal listening (e.g., participants evaluated stimuli presented through both CI and HA).
- Results
- General Condition: The results are presented clearly, but the figures (Figure 2 and Figure 3) need more descriptive captions. Specifically, indicate which comparisons were statistically significant in the captions to aid interpretation.
- Fitting Condition: Similarly, for Figure 3, ensure the caption highlights the key findings, such as the significant improvement in MOS with increased low-frequency gain for pop music with vocals.
- Statistical Reporting: When reporting ANOVA results, include effect sizes (e.g., η²) consistently for all analyses to provide a measure of practical significance. For instance, in the pop with vocals section, you reported η² = 0.776, but for other excerpts, it was omitted. Ensure completeness.
- Discussion
- Interpretation of Findings: The discussion effectively interprets the results, but expand on why the pop without vocals excerpt showed less sensitivity to manipulations. Consider suggesting that rhythmic elements might dominate for sparse instrumentation, which could be explored in future work.
- Clinical Implications: Strengthen the section on how these findings can be translated into clinical practice. For example, propose a step-by-step approach for audiologists to adjust HA settings based on music genre (e.g., increasing low-frequency gain for vocal-rich music). Also, discuss the potential for integrating these adjustments into fitting software (e.g., DSL or NAL-NL2 updates).
- Limitations: Address the sample size limitation more thoroughly. Discuss how the small n might affect statistical power and generalizability, and recommend future studies with larger, diverse cohorts (e.g., including users of different CI brands or those with varying musical backgrounds). Additionally, note that the acute nature of the trial does not account for long-term adaptation, which could be a focus for longitudinal research.
Minor Revisions Suggested
- Language and Grammar: Proofread the manuscript for minor errors. For example, in the "Music stimuli" section, "peace" should be corrected to "piece" (e.g., "excerpt of a single song").
- Data Presentation: In the results, consider adding a table summarizing the MOS values and statistical outcomes for all conditions and music excerpts to enhance clarity. This could supplement the figures.
- Ethical Considerations: Confirm that the ethics approval statement includes the approval date and reference number explicitly in the main text (currently it is in the document but ensure it is visible in the submitted version).
Author Response
Comments1:
The study demonstrates novelty in exploring HA parameter adjustments for music sound quality in bimodal CI users, a area with limited research. The integration of dynamic fitting modifications (e.g., compression settings and low-frequency gain) with subjective assessments via MUSHRA is a strength. However, the small sample size (n=13) and focus on a single CI manufacturer (Cochlear devices) may limit the generalizability of findings. I encourage author to acknowledge these limitations more explicitly and discuss their implications in the discussion section. Additionally, while the methodology is sound, some aspects of presentation and analysis could be refined to improve clarity and depth.
Response 1:
Thank you very much for your thoughtful comments. We have addressed these issues in the revised version of the manuscript and especially highlighted the potential limitations under the “conclusions” paragraph. We write: “A limitation is the small study sample that was mainly caused by the restricted choice of processor types, as explained above. This allowed a more controlled study setup on the one hand but limits the generalizability of the results on the other hand as it is not entirely clear how the (simulated) hearing aid would have interplayed with other electric stimulation strategies. For instance, given that electric stimulation could provide more low frequency information the effect of modifying acoustic low frequency gain might be weakened. Moreover, the limited sample size made it particularly difficult to perform a subgroup-analysis, e.g. based on the residual hearing of the participants.”
Specific Comments
- Abstract
- The abstract provides a good overview but could better highlight the key findings and their clinical significance. Specifically, emphasize that modifications to HA fitting (e.g., increased low-frequency gain for vocal-rich music) can significantly improve sound quality, as this is a central contribution. Also, mention the practical implications for personalized audiology care.
Response 2: Thank you, this is now highlighted in the abstract. We write: “Conclusions: The study demonstrates that HA fitting for bimodal CI users can be optimized beyond standard prescriptive rules to enhance music sound quality by increasing low-frequency gain particularly for vocal-rich pieces. Additionally, the testing method shows promise for clinical application, enabling individualized HA adjustments based on patient-specific listening preferences, hence fostering personalized audiology care.”
- Introduction
- The introduction sets the context well, but it would benefit from a more focused rationale on why music sound quality is a critical outcome measure for bimodal users. Consider citing recent studies (e.g., from the last 2-3 years) on music perception in CI users to update the literature review and strengthen the motivation. For example, work on neural correlates of music appreciation or patient-reported outcomes could be added.
Response 3: Recent literature had already been considered (i.e. [5, 6, 49]), and a study addressing neural correlates of music is added: “Additionally, the neural substrates of music perception [e.g., [7] Zattore & Salimpoor, 2013, https://doi.org/10.1073/pnas.130122811] might be crucial due to cortical changes and plasticity associated with hearing loss and cochlear implantation.”
- Methods
- Participants: The sample characteristics are described, but please provide more detail on the recruitment process and any inclusion/exclusion criteria related to musical experience or hearing loss etiology. This will help readers assess the representativeness of the cohort.
Response 4: We now write: “This restricted choice of devices was due to keep the methodological and especially the technical variability small. As this was the main inclusion criterion for the study, this resulted in a relatively large variability in terms of hearing loss etiology or musical experience. Residual hearing of the non-implanted ear was mainly present in the frequency range up to 1 kHz. The subjects were chosen to have a hearing loss of not larger than 85 dB HL in this frequency range as no benefit in music perception is expected for thresholds exceeding this value [23].”
We additionally highlight the limitations stemming from this choice more clearly in the conclusions section: “A limitation is the small study sample that was mainly caused by the restricted choice of processor types, as explained above. This allowed a more controlled study setup on the one hand but limits the generalizability of the results on the other hand as it is not entirely clear how the (simulated) hearing aid would have interplayed with other electric stimulation strategies. For instance, given that electric stimulation could provide more low frequency information the effect of modifying acoustic low frequency gain might be weakened.”
- Procedure: The use of the MUSHRA test with the drag-and-drop interface is well-explained. However, clarify how the "general condition" and "fitting condition" were randomized across participants to avoid order effects. Also, mention any training or familiarization sessions provided to participants to ensure reliable ratings.
Response 5: Half of the participants began with the “general condition” and the other half with the “fitting condition”. After explaining the MUSHRA procedure by giving examples for the rating procedure the measurements were started without further training. The reliability of the ratings was proved by the eGauge assessment. We write: “After loudness adjustment, the MUSHRA concept was explained by giving examples for the rating procedure. Half of the listeners began quality assessment in the general condition and half in the fitting condition. The presentation of the different music excerpts was pseudo-randomized.”
- Stimuli: The music excerpts are appropriately chosen, but include the spectral centroid values for each excerpt in the main text (not just in the methods) to better justify their selection for low-frequency emphasis. For instance, state: "The average spectral centroid was 683 Hz for classical, 613.8 Hz for pop with vocals, and 383 Hz for pop without vocals, ensuring sufficient low-frequency content."
Response 6: This was highlighted in the discussion section, where it now reads: “. First, in order to reflect manipulations of the HA that typically provides low frequency gain in bimodal CI users, a significant proportion of the spectrum should lie in this frequency area. This was ensured by determining the spectral centroid of the stimuli that was below 1 kHz in all cases, i.e. 683 Hz for classical, 613.8 Hz for pop with vocals, and 383 Hz for pop without vocals.”
- Figure 1: The MUSHRA interface is clearly depicted, but the caption could be expanded to explain how this interface was used in the context of bimodal listening (e.g., participants evaluated stimuli presented through both CI and HA).
Response 7: Thank you, this is included in the caption: “Schematic of the MUSHRA drag & drop interface. Each button depicted by a letter represents a sound sample. The participants evaluated these stimuli presented through both the CI and HA by positioning the buttons along the quality scale between 0 and 100.”
- Results
- General Condition: The results are presented clearly, but the figures (Figure 2 and Figure 3) need more descriptive captions. Specifically, indicate which comparisons were statistically significant in the captions to aid interpretation.
Response 8: We feel that including results of the statistics would be too comprehensive to be included in the captions and therefore refer to the text. However, we briefly explain the key findings and write: “Typically, DSL yields highest MOS and the anchor lowest MOS, depending on the music excerpt. Please refer to the text for the statistical analyses.”
- Fitting Condition: Similarly, for Figure 3, ensure the caption highlights the key findings, such as the significant improvement in MOS with increased low-frequency gain for pop music with vocals.
Response 9: Please see above. We write: “Increasing low frequency gain tends to yield higher MOS, depending on the music excerpt. Please refer to the text for the statistical analyses.”
- Statistical Reporting: When reporting ANOVA results, include effect sizes (e.g., η²) consistently for all analyses to provide a measure of practical significance. For instance, in the pop with vocals section, you reported η² = 0.776, but for other excerpts, it was omitted. Ensure completeness.
Response 10: Thank you, effect sizes are included.
- Discussion
- Interpretation of Findings: The discussion effectively interprets the results, but expand on why the pop without vocals excerpt showed less sensitivity to manipulations. Consider suggesting that rhythmic elements might dominate for sparse instrumentation, which could be explored in future work.
Response 11: Thank you, we have added: “In contrast, gain manipulations in sparsely instrumented music, such as the piece of pop without vocals, show a much smaller effect. We suspect that this is due to the superior role of rhythmic cues, which might dominate for this type of instrumentation.”
- Clinical Implications: Strengthen the section on how these findings can be translated into clinical practice. For example, propose a step-by-step approach for audiologists to adjust HA settings based on music genre (e.g., increasing low-frequency gain for vocal-rich music). Also, discuss the potential for integrating these adjustments into fitting software (e.g., DSL or NAL-NL2 updates).
Response 12: Thank you, this is included under “Conclusions”. We write: “This study showed that increasing the acoustic low-frequency gain has the potential to improve sound quality, especially for music with vocals. It thus points to possible clinical applications, e.g. by increasing hearing aid amplification for bimodal fitting, based on standard rules such as DSL, in a music-specific fashion, or by integrating higher low-frequency amplification into the fitting software in an application-specific manner.”
- Limitations: Address the sample size limitation more thoroughly. Discuss how the small n might affect statistical power and generalizability, and recommend future studies with larger, diverse cohorts (e.g., including users of different CI brands or those with varying musical backgrounds). Additionally, note that the acute nature of the trial does not account for long-term adaptation, which could be a focus for longitudinal research.
Response 13: This is also included under “Conclusions”. We write: “A limitation is the small study sample that was mainly caused by the restricted choice of processor types, as explained above. This allowed a more controlled study setup on the one hand but limits the generalizability of the results on the other hand as it is not entirely clear how the (simulated) hearing aid would have interplayed with other electric stimulation strategies. For instance, assuming that electric stimulation could provide more low frequency information, the effect of modifying acoustic low frequency gain may be weakened. Moreover, the limited sample size made it particularly difficult to perform a subgroup-analysis, e.g. based on the residual hearing of the participants.” and “Last not least, as an acute trial this study does not consider long-term adaptation effects, which could be a focus of longitudinal studies.”
Minor Revisions Suggested
- Language and Grammar: Proofread the manuscript for minor errors. For example, in the "Music stimuli" section, "peace" should be corrected to "piece" (e.g., "excerpt of a single song").
Response 14: Thank you, this was corrected.
- Data Presentation: In the results, consider adding a table summarizing the MOS values and statistical outcomes for all conditions and music excerpts to enhance clarity. This could supplement the figures.
Response 15: We feel that this may give somewhat redundant information but we can consider this as an appendix to the manuscript upon request.
- Ethical Considerations: Confirm that the ethics approval statement includes the approval date and reference number explicitly in the main text (currently it is in the document but ensure it is visible in the submitted version).
Response 16: Thank you, approval date is now included in the “Methods” section.
Round 2
Reviewer 1 Report
Comments and Suggestions for Authors
reviewed response to critiques, ok to accept